# ImmuneBuilder: Deep-Learning models for predicting the structures of immune proteins

Brennan Abanades [1], Wing Ki Wong [2], Fergus Boyles[1], Guy Georges[2], Alexander Bujotzek [2] & Charlotte M. Deane [1✉]

Immune receptor proteins play a key role in the immune system and have shown great promise as biotherapeutics. The structure of these proteins is critical for understanding their antigen binding properties. Here, we present ImmuneBuilder, a set of deep learning models trained to accurately predict the structure of antibodies (ABodyBuilder2), nanobodies (NanoBodyBuilder2) and T-Cell receptors (TCRBuilder2). We show that ImmuneBuilder generates structures with state of the art accuracy while being far faster than AlphaFold2. For example, on a benchmark of 34 recently solved antibodies, ABodyBuilder2 predicts CDR-H3 loops with an RMSD of 2.81Å, a 0.09Å improvement over AlphaFold-Multimer, while being over a hundred times faster. Similar results are also achieved for nanobodies, (NanoBody-Builder2 predicts CDR-H3 loops with an average RMSD of 2.89Å, a 0.55Å improvement over AlphaFold2) and TCRs. By predicting an ensemble of structures, ImmuneBuilder also gives an error estimate for every residue in its final prediction. ImmuneBuilder is made freely available, both to download (https://github.com/oxpig/ImmuneBuilder) and to use via our webserver (http://opig.stats.ox.ac.uk/webapps/newsabdab/sabpred). We also make available structural models for ~150 thousand non-redundant paired antibody sequences (https://doi.org/10.5281/zenodo.7258553).

[1] Department of Statistics, University of Oxford, Oxford, UK. [2] Large Molecule Research, Roche Pharma Research and Early Development, Roche Innovation Center Munich, Penzberg, Germany. ✉email: deane@stats.ox.ac.uk

The adaptive immune system in humans is effective at identifying and neutralising a wide range of pathogens. To achieve this, immune cells have developed antigen-specific proteins such as T-cell receptors (TCRs) or, in the case of B-cells, antibodies. While antibodies are capable of binding with great affinity and specificity to the surface of almost any antigen, TCRs target digested pieces of intracellular proteins that are presented on the cell surface by the major histocompatibility complex. Due to their key role in identifying a wide range of antigens, antibodies and TCRs have become proteins of particular interest for therapeutic development, with several TCR drugs in clinical trials[1] and over a hundred approved antibody drugs[2,3]. Nanobodies, single-domain antibodies naturally found in organisms such as camelids and sharks, have also received significant interest as therapeutics, with a recently accepted nanobody drug and a number undergoing clinical trials[4].

All three of these immune proteins are built up from immunoglobulin (Ig) domains with the binding site either sitting across two Ig domains in the case of antibodies (VH and VL) and TCRs (Vα and Vβ), or being found at the tip of one Ig domain, in the case of nanobodies.

The binding site of antibodies and TCRs is concentrated in six loops, three on each of the two Ig domains known collectively as the complementarity-determining regions (CDRs). In nanobodies, the binding site is concentrated in only three CDR loops on its single Ig domain. These CDR loops show variable length, composition and structure with the most variable being CDR-H3 in the case of antibodies and nanobodies[5]. This loop also tends to be the largest contributor to the binding site[6]. An example of the structure of an antibody variable domain, a TCR variable domain and a nanobody are shown in Fig. 1.

Despite the similarities in the global structure of antibodies, TCRs and nanobodies, their binding sites are known to have distinct properties and their CDRs have different length distributions as well as occupying distinct areas of structural space[7,8].

As with many proteins, the availability of sequence data far outstrips structural information[9–13], but structural information allows for a more in-depth understanding than studies focused on sequence alone[14]. For example, knowledge of CDR loop conformations has been used to help identify antibodies that bind to similar targets[15], while accurate knowledge of side chain atom placement can aid in identifying key interactions in antibody-antigen binding[16,17].

Experimental structure determination is time-consuming and expensive[18]. Computationally predicted structural models can be used to circumvent this problem. This is particularly the case for immune proteins, as next-generation sequencing of immune receptor repertoires is now routinely used in the study of the adaptive immune system[19,20]. These methods enable researchers to obtain millions of sequences per study, making structural analysis of this data a challenge. For example, Observed Antibody Space (OAS) contains over two billion antibody heavy chain sequences and is growing rapidly[9,10]. If this huge amount of sequence data is to even partially be analysed in terms of structure, rapid accurate methods for the prediction of antibody structures are required.

AlphaFold2 is a deep learning method that has revolutionised the field of computational protein structure prediction, achieving near experimental accuracy for a large number of single-chain proteins[21]. This was then extended to AlphaFold-Multimer to accurately predict protein complexes[22]. Many methods have followed from AlphaFold2 and AlphaFold-Multimer but these remain the de facto gold standard for single domains and complexes[23–25].

The AlphaFold2 model can be divided into two main steps: In the first step, the Evoformer module is used to extract evolutionary couplings from alignments of many protein sequences into information-rich embeddings. It then uses these embeddings in the structure module to predict the 3D structure of a given protein sequence.

Structure prediction methods specific to a certain class of protein tend to outperform more general methods[26,27]. By using knowledge specific to a type of protein, they can easily predict the conserved regions in that protein allowing greater focus on harder details. For example, DeepH3 was shown to outperform TrRosetta on antibodies[28,29], while Nanonet obtains results of similar accuracy to AlphaFold2 on nanobodies with a far simpler architecture[30]. More recent examples of this are IgFold[25] and EquiFold[31], where the authors trained antibody-specific models that predict structures of comparable accuracy to AlphaFold-Multimer.

In this paper, we present ImmuneBuilder, a set of deep learning models developed to predict the structure of proteins of the immune system. By training on specific protein types, we are able to create rapid accurate models, enabling ImmuneBuilder to be routinely used on large sequence data sets. We have built three models, ABodyBuilder2, an antibody-specific model, NanoBodyBuilder2, a nanobody-specific model and TCRBuilder2 a TCR-specific model. We show that these methods perform at least as well as state-of-the-art methods for their respective protein types while predicting

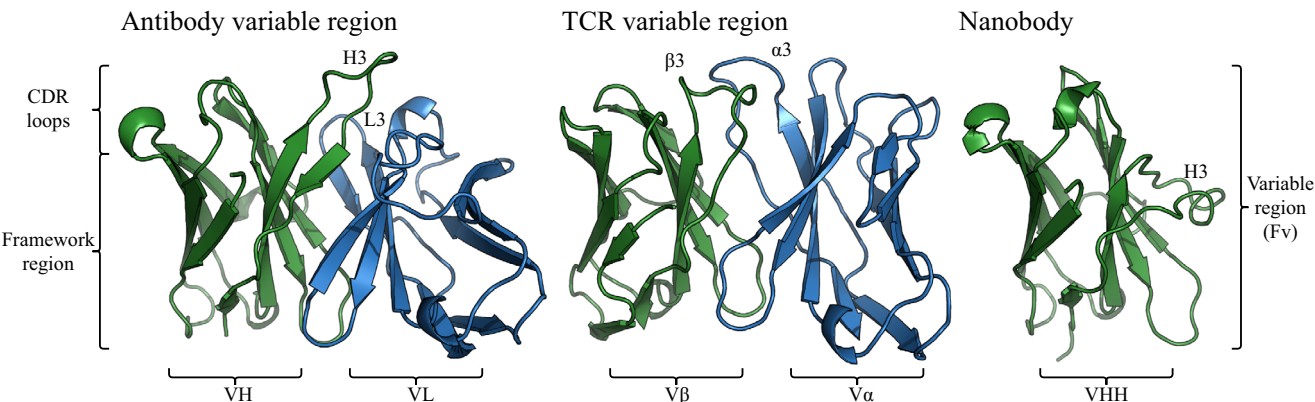

**Fig. 1 Structural representation of an antibody variable domain (PDB code 1GIG), a TCR variable domain (PDB code 7SU9) and a nanobody (PDB code 4LAJ) with labelled regions.** The heavy and beta chains are shown in green while the light and alpha chains are blue. As these structures show, antibodies and TCRs are structurally similar and the nanobody is also similar to an individual chain of an antibody or TCR. However, their CDR loops occupy distinct areas of structural space.

structures in a fraction of the time. We also demonstrate that these methods both accurately predict details of the structure and create physically and biologically sensible structures.

The three ImmuneBuilder models are made freely available for download and as web servers.

## Results

Throughout the results section we will focus on the results for ABodyBuilder2 (AB2) on antibodies with the results for Nano-BodyBuilder2 (nanobodies) and TCRBuilder2 (TCRs) discussed in Supplementary Notes 1 and 2. All three methods show qualitatively similar results.

We compare ABodyBuilder2 to several other methods. These methods are a homology modelling method (the original version of ABodyBuilder[32] (ABB)), one general protein structure prediction method (AlphaFold-Multimer[22] (AFM)), and three antibody-specific methods (ABlooper[33] (ABL), IgFold[25] (IgF) and EquiFold[31] (EqF)). As a benchmark, we selected a non-redundant set of 34 antibody structures recently added to SAbDAb[11,13] (see methods). This was done so none of the antibody structures in the benchmark would have been seen during training for any of the benchmarked methods. To give a complete picture of how these methods perform, we carryout a comprehensive benchmark using five different measures. Figure 2 shows an example of a prediction by ABodyBuilder2, highlighting important aspects of structural modelling.

**Accuracy of prediction**. To measure how accurately the backbone atoms are predicted, the RMSD between predicted and true structures for each antibody region was compared. The RMSD for each CDR and framework is computed by aligning each protein chain to the crystal structure and then calculating the RMSD between the $C_\alpha$, N, C and $C_\beta$ atoms. Regions are defined using the IMGT numbering scheme[34]. The results of this analysis are shown in Table 1.

The experimental error in protein structures generated via X-ray crystallography has been estimated to be around 0.6Å for regions with organised secondary structures (such as the antibody frameworks) and around 1Å for protein loops[35]. On average, the predicted structures for most of the antibody regions using any method have errors within the range of what would be expected from experimentally resolved crystal structures. The exception to this is CDR-H3, where all methods make the worst predictions.

ABodyBuilder2 and AlphaFold-Multimer are the most accurate methods at predicting the structure of CDR-H3 (RMSD of 2.81 Å and 2.90 Å, respectively). EquiFold, IgFold and ABlooper generate structures with CDR-H3 loops around 10% less accurate than ABodyBuilder2 and AlphaFold-Multimer. The worst method for predicting CDR-H3 loops is the original version of ABodyBuilder, showcasing how deep learning has improved our ability to model. Supplementary Note 4 explores how the accuracy of ABodyBuilder2 for CDR-H3 prediction correlates to the maximum sequence identity in the training set. A comparison of the CDR-H3 RMSD for each individual structure in the test set between each pair of methods is shown in Supplementary Fig. 4.

Tables 2 and 3 show how accurate TCRBuilder2 and Nano-BodyBuilder2 are at predicting the position of atoms in the backbone. We compare them to homology modelling methods

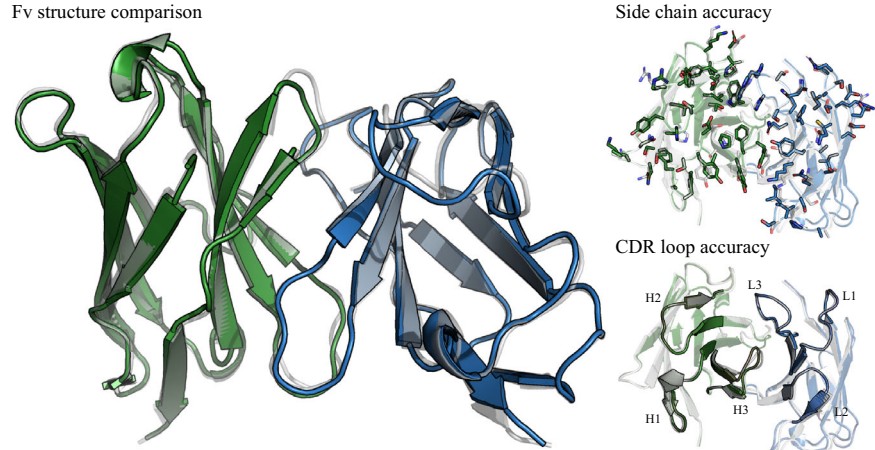

**Fig. 2 Example of an antibody structure predicted with ABodyBuilder2.** The heavy chain is shown in green, the light chain in blue and the crystal structure in white. The figure on the left shows the overall Fv structure, demonstrating how ABodyBuilder2 accurately predicts the relative orientation between the heavy and light chain. The figures on the right focus on the structure of CDR loops. The top image showcases the accuracy of ABodyBuilder2 at modelling side chain atoms and the bottom image shows its accuracy at predicting the backbone of CDR loops.

**Table 1 Comparison between ABodyBuilder, ABlooper, IgFold, EquiFold, AlphaFold-Multimer and ABodyBuilder2 at predicting the backbone atoms for each antibody chain.**

| Method | CDR-H1 | CDR-H2 | CDR-H3 | Fw-H | CDR-L1 | CDR-L2 | CDR-L3 | Fw-L |
|---|---|---|---|---|---|---|---|---|
| ABodyBuilder (ABB) | 1.53 | 1.09 | 3.46 | 0.65 | 0.71 | 0.55 | 1.18 | 0.59 |
| ABlooper (ABL) | 1.18 | 0.96 | 3.34 | 0.63 | 0.78 | 0.63 | 1.08 | 0.61 |
| IgFold (IgF) | 0.86 | 0.77 | 3.28 | 0.58 | 0.55 | 0.43 | 1.12 | 0.60 |
| EquiFold (EqF) | 0.86 | 0.80 | 3.29 | 0.56 | 0.47 | 0.41 | 0.93 | **0.54** |
| AlphaFold-M (AFM) | 0.86 | **0.68** | 2.90 | 0.55 | 0.47 | **0.40** | **0.83** | **0.54** |
| ABodyBuilder2 (AB2) | **0.85** | 0.78 | **2.81** | **0.54** | **0.46** | 0.44 | 0.87 | 0.57 |

The mean RMSD to the crystal structure across the test set for each of the six CDRs and framework (Fw) is shown. RMSDs are given in Angstroms (Å). The best-performing method for each metric is highlighted in bold. The individual RMSDs for each method for each CDR are given in Supplementary Data 1.

**Table 2 Comparison between TCRBuilder, RepertoireBuilder, AlphaFold-Multimer, ABodyBuilder2 and TCRBuilder2 at predicting the backbone atoms for each TCR chain.**

| Method | CDR-α1 | CDR-α2 | CDR-α3 | Fw-α | CDR-β1 | CDR-β2 | CDR-β3 | Fw-β |
|---|---|---|---|---|---|---|---|---|
| TCRBuilder | 1.60 | 1.31 | 2.89 | 0.87 | 0.99 | 0.90 | 3.12 | 0.81 |
| RepertoireBuilder | 1.35 | 1.00 | 2.64 | 0.75 | 0.86 | 1.59 | 2.77 | 1.05 |
| AlphaFold-M | **1.25** | 0.96 | **1.84** | **0.69** | 0.75 | 0.65 | 1.94 | 0.82 |
| ABodyBuilder2 | 3.49 | 6.57 | 3.14 | 2.89 | 3.27 | 3.77 | 3.48 | 3.65 |
| TCRBuilder2 | 1.34 | **0.93** | 1.85 | 0.90 | **0.74** | **0.63** | **1.93** | **0.67** |

The mean RMSD to the crystal structure across the TCR test set for each of the six CDRs and framework (Fw) is shown. RMSDs are given in Angstroms (Å). The best-performing method for each metric is highlighted in bold. The individual RMSDs for each method for each CDR are given in Supplementary Data 2.

**Table 3 Comparison between ABodyBuilder, MOE, AlphaFold2 and NanoBodyBuilder2 at predicting the backbone atoms of nanobodies.**

| Method | CDR1 | CDR2 | CDR3 | Fw |
|---|---|---|---|---|
| ABodyBuilder | 2.96 | 2.08 | 5.08 | 1.09 |
| MOE | 2.67 | 1.99 | 4.90 | 1.19 |
| AlphaFold2 | 2.08 | **1.35** | 3.44 | 0.82 |
| NanoBodyBuilder2 | **1.98** | 1.37 | **2.89** | **0.79** |

The mean RMSD to the crystal structure across the nanobody test set for each of the three CDRs and framework (Fw) is shown. RMSDs are given in Angstroms (Å). The best-performing method for each metric is highlighted in bold. The individual RMSDs for each method for each CDR are given in Supplementary Data 3.

**Table 5 Comparison of surface accuracy for ABodyBuilder (ABB), ABlooper (ABL), IgFold (IgF), EquiFold (EqF), AlphaFold-Multimer (AFM) and ABodyBuilder2 (AB2).**

| Method | $\chi 1$ | $\chi 2$ | $\chi 3$ | $\chi 4$ | E/B |
|---|---|---|---|---|---|
| ABB | 0.81 | 0.77 | **0.63** | 0.56 | 0.92 |
| ABL | 0.75 | 0.70 | 0.60 | 0.53 | 0.90 |
| IgF | 0.77 | 0.66 | 0.54 | 0.52 | 0.91 |
| EqF | 0.84 | 0.71 | 0.61 | 0.55 | **0.94** |
| AFM | **0.85** | 0.77 | 0.59 | **0.58** | 0.91 |
| AB2 | **0.85** | **0.78** | **0.63** | 0.52 | 0.91 |

Values shown are percentages representing the accuracy when modelling each of the first four torsion angles of the side chain ($\chi$) if they exist. E/B gives the accuracy at predicting whether a residue is exposed or buried. The best-performing method for each metric is highlighted in bold.

**Table 4 Comparison of VH-VL orientation between ABodyBuilder (ABB), ABlooper (ABL), IgFold (IgF), EquiFold (EqF), AlphaFold-Multimer (AFM) and ABodyBuilder2 (AB2).**

| Method | HL | HC1 | LC1 | HC2 | LC2 | dc |
|---|---|---|---|---|---|---|
| Xtal | 1.18 | 0.48 | 0.75 | 0.62 | 0.80 | 0.11 |
| ABB | 0.83 | 1.09 | 0.82 | 1.81 | 0.90 | **0.10** |
| ABL | 0.80 | 0.97 | 0.83 | 1.70 | 0.90 | 0.12 |
| IgF | **0.63** | 0.91 | 0.71 | 1.40 | 0.74 | 0.14 |
| EqF | 0.64 | 0.98 | **0.64** | 1.68 | 0.72 | 0.11 |
| AFM | 0.67 | **0.74** | 0.69 | 1.45 | 0.63 | 0.11 |
| AB2 | 0.64 | 0.90 | 0.66 | **1.37** | **0.61** | 0.12 |

Values shown were calculated using ABangle[38]. HL, HC1, LC1, HC2, LC2 and dc are defined in Supplementary Fig. 2 and ref. [38]. The error in the angles HL, HC1, LC1, HC2 and LC2 is shown in degrees with the error in the distance dc given in Angstroms (Å). The average standard deviation observed in antibodies experimentally resolved over five times is shown for comparison (Xtal). The best-performing method for each metric is highlighted in bold.

(RepertoireBuilder[36] and TCRBuilder[26] for TCR modelling, and MOE[37] and ABodyBuilder[32] for nanobody modelling) and machine learning methods AlphaFold-Multimer[22] for TCRs and AlphaFold2[21] for nanobodies. Supplementary Fig. 1 provides a visual representation of the potential differences in the predicted conformation of the CDR-H3 region using NanoBody-Builder2 and ABodyBuilder2, even when they correspond to the same sequence. Full results for TCRBuilder2 and NanoBody-Builder2 and details on their respective test sets are given in Supplementary Notes 1 and 2.

**Heavy and light chain packing**. As described in the introduction, in antibodies the binding site sits across the heavy and light chain variable regions (VH and VL). With half of the CDRs on each chain, the relative VH-VL orientation can have an impact on the structure of the binding site.

To quantify how accurate each method is at predicting the relative orientation between chains, in Table 4 we show the average absolute error in the five angles (Hl, HC1, HC2, LC1, LC2) and distance (dc) that fully characterise VH-VL orientation[38]. A brief description of how these values are defined is given in Supplementary Note 3, for a more complete description see ref. [38]. The results for TCR domains are given in Supplementary Table 2.

As an upper bound for the accuracy of predicted structures, the average standard deviation of the VH-VL orientation measurements in 55 antibodies with structures resolved over five times is shown in Table 4. In the original study[38], the vector dc was chosen as an axis as it was found to be the most conserved amongst antibody structures. All of the benchmarked methods predict this distance with very high accuracy. All methods are also accurate at predicting the angles, with small errors with respect to what is observed in experiments. However, small deviations in these angles will still have an impact on the structure of the binding site. ABodyBuilder2 is on average the most accurate method at heavy and light chain packing by a small margin.

**Side chain and chemical surface accuracy**. During binding, an antigen will mostly form interactions via side chain atoms on the surface. Therefore to be able to study antigen binding, predicted antibody structures must accurately model the position of side chain atoms and whether they are exposed on the surface or buried. To benchmark the accuracy of side-chain modelling we use a method similar to ref. [39], where a side-chain torsion angle is considered correct if it is within 40 degrees of the true conformation. The original implementation of ABodyBuilder will occasionally fail to predict a side chain, this is treated as an incorrect prediction. A residue is labelled as buried if its relative solvent accessibility (calculated as described in ref. [40]) is below 7.5%. The results of this analysis are given in Table 5 for ABodyBuilder2 and in Supplementary Tables 1 and 2 for Nanobodies and TCRs respectively.

As ABlooper and IgFold are deep learning methods that only predict the backbone (leaving side chain prediction to OpenMM[41] and Rosetta[42], respectively), it is perhaps not surprising that they are the least accurate at modelling the chemical surface. EquiFold,

AlphaFold-Multimer and ABodyBuilder2, all of which output all-atom structures, predict the $\chi 1$ and $\chi 2$ side chain atoms with high accuracy while struggling to model longer side chains. The original implementation of ABodyBuilder predicts side chains with comparable accuracy to AlphaFold-Multimer and ABodyBuilder2. All methods are highly accurate at predicting whether a residue is exposed or buried, EquiFold is the most accurate.

**Physical plausibility and accurate stereochemistry**. Although deep learning models are trained on crystal structures, they will occasionally predict conformations that are very rare or do not occur in nature. We next check for the presence of steric clashes, cis-peptide bonds, D-amino acids, or bonds with nonphysical lengths in the models generated by each method. For bond lengths, only the peptide bond is checked as all other bond lengths are fixed to their literature value in all benchmarked methods but ABlooper and EquiFold.

ABodyBuilder2 and AlphaFold-Multimer both generate structures of comparable quality to experimentally resolved ones, whereas IgFold appears to generate a number of cis-peptide bonds and clashes even after being refined with Rosetta[42]. EquiFold does not use an energy-based method to refine its predicted structures and hence all of the structures it generates are unphysical. This shows that a refinement step is still necessary to ensure structures generated by deep learning-based methods are realistic (Table 6).

The results for TCRs and nanobodies are shown in Supplementary Tables 1 and 2, respectively.

**Computational cost**. The original version of AlphaFold-Multimer is by far the most computationally expensive of the benchmarked methods. It requires over one terabyte of sequence data and takes around three hours to generate one structure when run on five CPUs. Large speed-ups can be obtained by reducing the size of the sequence database, using faster sequence alignment algorithms, or using GPUs[43,44]. Even with these modifications, it takes around thirty minutes on a GPU to generate a single structure. All other methods benchmarked can be run on five CPUs in under a minute, with the fastest being EquiFold due to its lack of a refinement step. This makes them all well suited for high throughput structural modelling of next-generation sequencing data. ABodyBuilder2 can also be sped up significantly by using a GPU taking around five seconds to generate an antibody structure on a Tesla P100.

**Error estimation**. ABodyBuilder2 predicts four structures for each antibody. We found that the diversity between predictions,

as in ABlooper, can be used to estimate the uncertainty in the final prediction. If the structures predicted by all four models disagree in a region then the final prediction for this region is likely to be incorrect. This allows ABodyBuilder2 to give a confidence score for each residue that can be used to filter for incorrectly modelled structures. In Fig. 3 we show how the root mean squared predicted error for CDR-H3 residues correlates with CDR-H3 RMSD.

A low predicted error does not necessarily indicate an accurate structure. However, a high predicted error works as a good filter for removing inaccurate models. For example, if a predicted error cut-off of around 1 Å is set for the current benchmark, it would remove five structures with an average RMSD of 4.46 Å. The average CDR-H3 RMSD for the remaining set would then be 2.53 Å.

## Discussion

We present ImmuneBuilder, a set of three open-source and freely available tools for modelling immune proteins capable of rapidly generating accurate antibody, TCR, and nanobody structures. ImmuneBuilder can produce structures of antibodies and TCRs with accuracy comparable to AlphaFold-Multimer while being over a hundred times faster and without the need for large sequence databases or multiple sequence alignments. ABody-Builder2 is shown to be the most accurate of the antibody-specific tools and the only one to consistently predict structures with correct stereochemistry.

The comparison with homology modelling methods, such as ABodyBuilder, shows the benefits that deep learning has brought to the field of antibody structure prediction. However, all methods still struggle to accurately predict the conformation of CDR-H3, suggesting that models capable of predicting multiple conformations may be required to accurately capture this loop. Deep learning methods also still struggle to consistently predict physically plausible structures. This challenge can be addressed by using physics-based methods, such as restrained energy minimisation, but for fast methods like ABodyBuilder2 this greatly increases computational cost.

By measuring the variability between predictions, Immune-Builder is able to provide an error estimate for each residue. In combination with its prediction speed and accuracy, the ability to filter for incorrect models makes it a useful tool for incorporating

**Table 6 Quality check for models generated using ABodyBuilder (ABB), ABlooper (ABL), IgFold (IgF), EquiFold (EqF), AlphaFold-Multimer (AFM) and ABodyBuilder2 (AB2).**

| Method | Peptide bond | D-amino acid | Cis-bond | Clash |
|---|---|---|---|---|
| Xtal | 0 | 0 | 0 | 0 |
| ABB | 19 | 0 | 3 | 9 |
| ABL* | 1 | 0 | 1 | 1 |
| IgF | 0 | 0 | 51 | 10 |
| EqF | 271 | 4 | 2 | 765 |
| AFM | 0 | 0 | 0 | 0 |
| AB2 | 0 | 0 | 0 | 0 |

The errors observed in experimentally resolved crystal structures (Xtal) are also shown for comparison. A peptide bond length is considered to be incorrect if it is more than 0.1 Å away from its literature value. Peptide bonds including a proline were not included in the calculation of cis-isomers. Two non-bonded heavy atoms are considered to be clashing when they are closer than 0.63 times their Van-der-Waals radius[39]. (*) Calculated using the latest version of ABlooper, updated since publication to reduce the number of D-amino acids and cis-isomers.

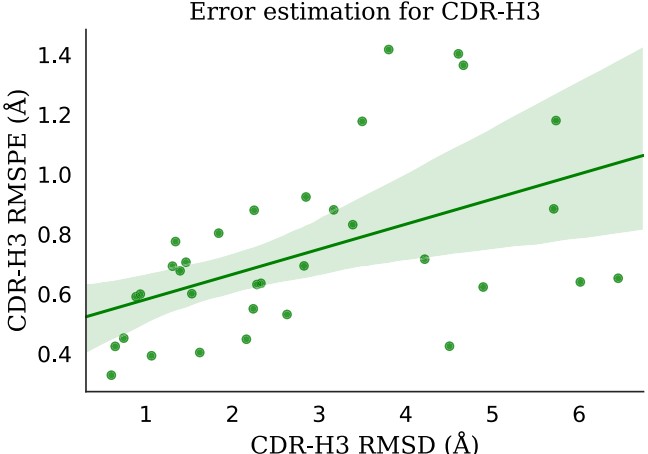

**Fig. 3 Scatter plot showing the CDR-H3 RMSD against the root mean squared predicted error for all structures in the benchmark.** The line shown is the best fit with the 95% confidence interval shown as the shaded area around it. The data used to generate this plot is given in Supplementary Data 4.

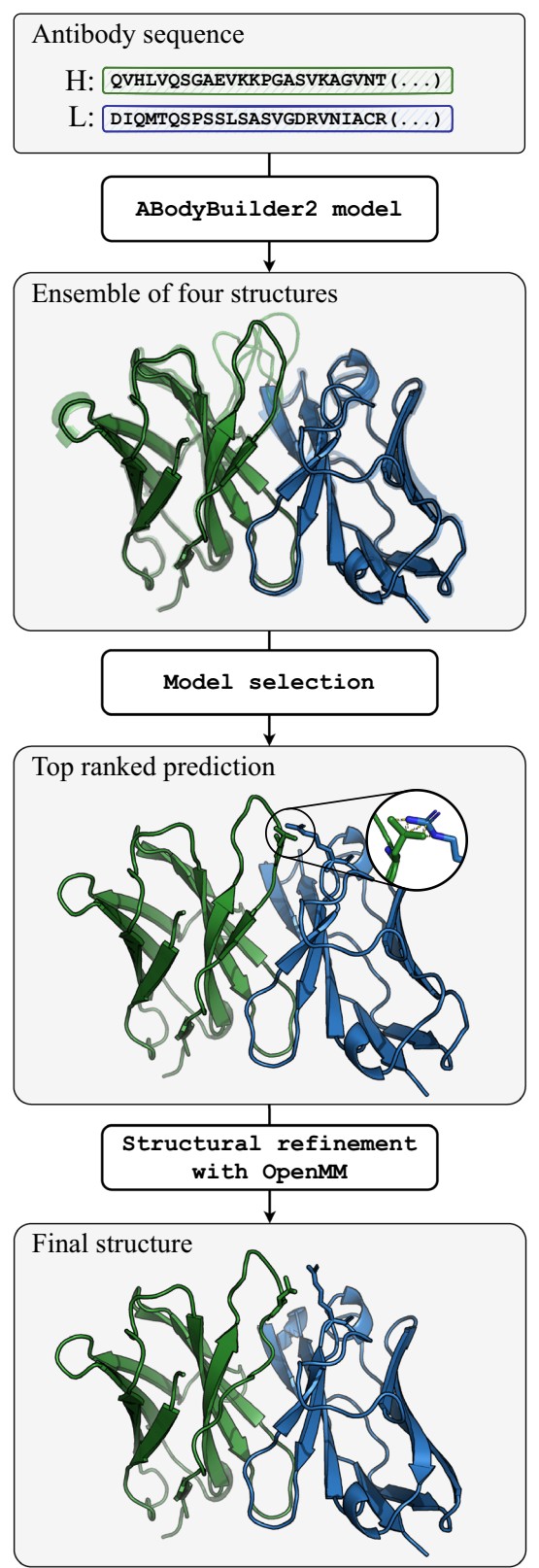

**Fig. 4 Pipeline used to predict structures by ABodyBuilder2.** First, the heavy and light chain sequences are fed into four separate deep-learning models to predict an ensemble of structures. The closest structure to the average is then selected and refined using OpenMM[41] to remove clashes and other stereo-chemical errors. The same pipeline is used for NanoBodyBuilder2 and TCRBuilder2.

structural information into data from next-generation sequencing experiments.

To further demonstrate the usefulness of ImmuneBuilder, we predicted the structure of around 148 thousand non-redundant paired antibody sequences from OAS[10] and make these freely available at (https://doi.org/10.5281/zenodo.7258553).

## Methods

In the methods section we describe in detail the data and models for creating ABodyBuilder2. An overview of the steps used to predict an antibody structure with ABodyBuilder2 is shown in Fig. 4. Details for NanoBodyBuilder2 and TCRBuilder2 are given in Supplementary Notes 1 and 2.

**Data**. The data used to train, test, and validate ABodyBuilder2 was extracted from SAbDab[11], a database containing all antibody structures in the PDB[45]. The training data was extracted on the 31st of July 2021 resulting in a total of 7084 structures. Filters were used to ensure structures in the training data had both the VH and VL chains, were not missing residues other than at the start and end of the chain, and had a resolution of 3.5 Å or better. Structures with the same amino acid sequence were kept in the training data to expose the model to antibodies with multiple conformations. As a validation set, we used the set of 49 antibodies in the Rosetta Antibody Benchmark. Structures with the same sequence as antibodies in the validation set were removed from the training set.

For the test data, we extracted all PDB files containing antibody Fv structures in SAbDab added between the 1st of August 2021 and the 1st of June 2022. Only crystal structures resolved by X-ray diffraction and with a resolution better than 2.3 Å were kept. A set of non-redundant Fvs were then selected from these and further filtered to remove antibodies with CDR-H3s longer than 22 amino acids and structures with missing residues. Finally, it was ensured that there were no structures with the same sequence in the test, training, and validation sets. This resulted in the set of 34 Fvs that was used to benchmark ABodyBuilder2 against other methods. A comparison of the maximum sequence identity to the training set against CDR-H3 RMSD for each Fv in the test set is shown in Supplementary Fig. 3. A full list of PDB codes for structures used in the training, validation and test set is given at https://github.com/oxpig/ImmuneBuilder.

**Deep learning architecture**. The architecture of the deep learning model behind ABodyBuilder2 is an antibody-specific version of the structure module in AlphaFold-Multimer with several modifications. Residues are treated as rigid bodies, each one defined by a 3D point in space and a matrix representing its orientation. The input node features are a one-hot encoded representation of the sequence and the input edge features are relative positional encodings. At the start, all residues are set at the origin with the same orientation.

The model is composed of eight update blocks that run sequentially. At every iteration, the node features are first updated in a structurally aware way using the Invariant Point Attention layer, and then residue coordinates and orientations are updated using the Backbone Update layer. For further details on how these layers work, see the original AlphaFold2 paper[21]. Finally, torsion angles for each residue are predicted from node features and are then used to reconstruct an all-atom structure using hard-coded rules. Unlike AlphaFold-Multimer, all blocks have their own weights.

The main term in the loss function is the Frame Aligned Point Error (FAPE) loss, which quantifies how structurally similar the true and predicted structures are in the local reference frame of each residue. For details see ref. [21]. In AlphaFold2, the FAPE loss is clamped at 10 Å focusing on correctly placing residues relative to those closest to it. Similar to AlphaFold-Multimer, a modified version of FAPE loss is used for ABodyBuilder2 in which more focus is given to correctly placing CDR residues relative to the framework. This is achieved by clamping the FAPE loss at 30 Å when it is calculated between framework and CDR residues and at 10 Å otherwise. The final loss term is a sum of the average backbone FAPE loss after every backbone update and the full atom FAPE loss from the final structure.

As is done in AlphaFold2, a structural violation term is added to the loss function. This penalises nonphysical conformations with a term for bond angles, bond distances, and clashing heavy atoms. In our models, this term was reduced by an order of magnitude with respect to AlphaFold2 as this was found to slightly improve prediction accuracy without significantly harming the physicality of the final prediction. Finally, the side-chain and backbone torsion angle loss from AlphaFold2 is also used.

Each model was trained in two stages. In the first stage, the structural violations term of the loss function was set to zero and a dropout of 10% was used. The RAdam optimiser[46] was used with a cosine annealing scheduler with warm restarts every 50 epochs, learning rates between 1e-3 and 1e-4, and a weight decay of 1e-3. For the second stage, the structural violations loss is added and dropout is set to zero. RAdam is also used for this stage with a fixed learning rate of 1e-4 and weight decay of 1e-3. To aid with stability, the norm of gradients is clipped to a value of 0.1 in the second stage of training. For both stages, a batch size of 64 is used and

training is stopped if there was no improvement in the validation set after 100 epochs. On average, training took around four weeks for each model on a single GPU.

**Model selection**. ABodyBuilder2 is composed of four deep-learning models trained independently to predict antibody structures. To select the best prediction, we align all predicted structures and choose as the final prediction the closest one to the average. This reduces the method's sensitivity to small fluctuations in the training set. It also results in a small improvement in prediction accuracy.

**Structural refinement**. Although the models are encouraged to predict physically plausible structures during training, they will occasionally produce structures with steric clashes, incorrect peptide bond lengths, or cis-peptide bonds. A restrained energy minimisation procedure with OpenMM is used to resolve these issues. The AMBER14 protein force field[47] with an added harmonic force term to keep the heavy atoms of the backbone close to their original positions is used. In the rare case when two side chain atoms are predicted by the model to be within 0.2 Å of each other, the clashing side chains are deleted and remodelled using pdbfixer[41]. AMBER14 does not explicitly consider chirality, so when the predicted structure contains peptide bonds in the cis configuration, an additional force is added to flip their torsion angles into the trans configuration.

By design, the ABodyBuilder2 deep learning model will always generate amino acids in their L-stereoisomeric form. However, it was found that during energy minimisation residues are occasionally flipped into their D-stereoisomer. To fix this, a method similar to that in ref. [48] is used. First, the chirality at the carbon alpha centre of each D-stereoisomeric residue is fixed by flipping the hydrogen atom. The structure is then relaxed keeping the flipped hydrogen atoms in place before a final minimisation.

**Benchmarked methods**. We compared ABodyBuilder2 to five other methods: AlphaFold-Multimer, EquiFold, IgFold, the original version of ABodyBuilder and ABlooper. AlphaFold-Multimer was run using the freely available version of code[22]. It was run using the weights from version 2.2 and without the use of templates. The effect of templates on antibody structure prediction is shown in Supplementary Table 3. This generated 25 structures per antibody out of which the top-ranked was selected for the benchmark. The public version of their respective code bases (as of December 15th) was used to generate EquiFold[31] and IgFold[25] models. As in their paper, Rosetta[42] is used to minimise IgFold models. The original version of ABodyBuilder[32] was run by using the SAbBox Singularity container (https://process.innovation.ox.ac.uk/software/p/20120-a/sabbox-singularity-platform—academic-use/1) from July 2022 excluding all templates with a sequence identity of 99% or higher. ABlooper[33] (version 1.1.2) was run to remodel the CDR loops from the ABodyBuilder predictions. Structures generated by all methods were numbered using ANARCI[49].

**Reporting summary**. Further information on research design is available in the Nature Portfolio Reporting Summary linked to this article.

## Data availability

The data used to generate the ImmuneBuilder models was extracted from public repositories such as SAbDab[11] and STCRDab[12]. All data generated from this study is available in the public repository located at https://doi.org/10.5281/zenodo.7258553.

## Code availability

Source code for the ImmuneBuilder models, trained weights and inference script are available under an open-source license at https://github.com/oxpig/ImmuneBuilder.

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

## Acknowledgements

This work was funded by the Engineering and Physical Sciences Research Council (EPSRC) with grant number EP/S024093/1 and Roche.

## Author contributions

B.A. and C.M.D. conceived the project and designed the study with input from all authors. B.A. designed and implemented the deep learning model. B.A. and W.K.W. ran ablation studies to optimise the model architecture and training procedure. B.A. trained the final models and implemented the final version of the code. B.A. compared ImmuneBuilder against other methods and compiled the results. F.B. made a web-server to run the model and visualise predictions. B.A. and C.M.D. wrote the manuscript with input from all authors. C.M.D., W.K.W., G.G., and A.B. supervised the project.

## Competing interests

The authors declare no competing interests.
