## [Peer Review File · Communications Biology]

Reviewers' comments:

Reviewer #1 (Remarks to the Author):

This study reports the development and benchmarking of ImmuneBuilder, which are deep learning protocols to model antibody, nanobody, and TCR structures. These algorithms are much faster than AlphaFold-Multimer, at least as accurate, and are available to the community via web server interfaces and Github. Additionally, sets of antibody structural models are made available by the authors on Zenodo. There are several aspects of the presentation of the results and methods that should be addressed, as noted in the comments below, but overall this is a very nice study that should be of considerable interest to the research community.

1. The authors note in the Results section (page 3): "ABodyBuilder2 is the most accurate method at predicting the structure of CDR-H3 (RMSD of 2.81 Å), closely followed by AlphaFold-Multimer (RMSD of 2.90 Å)." It is not clear whether this 0.09 Å improvement in mean RMSD is meaningful enough to say whether ABodyBuilder2 is truly better than AlphaFold-Multimer, or essentially the same in terms of accuracy. The authors should perform a statistical test (e.g. Wilcoxon signed rank on the individual CDRH3 RMSDs) to show whether the RMSDs from ABodyBuilder2 show a statistically significant improvement over AlphaFold-Multimer. Such a test would help to support any statements of superiority by the authors. If there is no significant difference or improvement, the authors should note that as well.
2. The authors do not seem to give the individual CDR RMSDs for their antibody, TCR, and nanobody test sets. Readers or users may want to know what the actual RMSDs were, versus just the reported mean values. This information should be provided as supplemental tables.
3. While it is understandable that the authors focus on antibody modeling performance in the main text and results, it would be helpful for readers to also be able to view the main performance results for TCRs and nanobodies, without needing to refer to the supplemental information. The two tables reporting accuracy performance (RMSDs) for nanobodies and TCRs should be moved from supplemental to the main text and results.
4. For the antibody results, the authors provide comparative results with available methods aside from AlphaFold-Multimer (i.e. IgFold and EquiFold), which is great. However, it would be nice if the authors likewise compared their nanobody and TCR modeling performance against at least one more method that was developed outside of their group and currently available to the community. That could include NanoNet, which was noted by the authors, and for TCRs, results for RepertoireBuilder or TCRmodel could be included, for example. This would help to provide a more comprehensive view on the reported performance for the test sets used by the authors.
5. The Methods section (page 7) notes that "Finally, it was ensured that there were no structures with the same sequence in the test, training, and validation sets." As written, it seems possible that two antibody structures with only one amino acid mismatch would possibly be in the training and test sets. The authors should more clearly note their nonredundancy criteria to ensure lack of train/test overlap (e.g. 99% identity, 95% identity) so that this is more clear to readers. If their training and test sets truly have a relatively permissive identity cutoff (e.g. 99% identity threshold), the authors should comment on why this would not be a concern.
6. It is reported by the authors that AlphaFold-Multimer was run without the use of templates for the antibody modeling. It is not clear why templates were not used, particularly as it is possible to set a template date cutoff to avoid overlap with the test set, and it is theoretically possible that templates may help with the AlphaFold-Multimer modeling. The authors should run AlphaFold-Multimer for their antibody test set, allowing templates with the appropriate date cutoff to avoid recent structures that would overlap with their test set, and report that performance. This would provide a more useful comparator for readers, who would potentially run AlphaFold-Multimer's default protocol, which does include templates, on prospective modeling targets.
7. On the ImmuneBuilder site, a TCR model was successfully generated (which is nice), but selecting "Prediction error" under "Annotation options" for the TCR model made the model

disappear in the viewer. However, the prediction error is visible for the sample TCR output on the web server. The authors should check this, to make sure that that aspect of the model output page is functioning properly, and if not, it should be fixed.

8. Page 2. "sitting between two Ig domains" does not seem to reflect the antigen binding of TCRs and antibodies, as the CDR loops are not actually between the two chains (e.g. within the VH-VL interface), they just form a contiguous surface formed by both domains. Thus the authors should consider re-wording that sentence.

9. The authors should include a statement or paragraph noting any shortcomings, failures, or areas of potential improvement for these methods.

Reviewer #2 (Remarks to the Author):

The manuscript presents a set of deep learning models for predicting structures of antibodies, nanobodies and T-cell receptors. The architecture of the models is inspired by AlphaFold2, adjustments to immune receptors are made. The method is trained and validated using existing structures and achieving state-of-the-art performance. Overall, the method is an excellent addition to the existing structure prediction approaches for immune receptors.

The authors provide a notebook for predicting the structures using a colab notebook and a webserver. The source code is also available, although I did not find the training part.

Comments:

Presentations of the results:

Only mean RMSD values are reported for the test set. It would be useful to add plots with distributions, such as swarmplots, especially for CDR3.

Do the different methods fail on the same test cases or different ones? Is it possible to include scatterplots of CDR3 RMSDs for comparison to additional methods? Highlight cases where AbodyBuilder2 is successful and where it fails?

Splitting the data to train/test – only identical sequences in the validation and train sets were removed from the training set. This is suboptimal and most likely test set sequences that have immune receptors with high sequence identity in the training set are modeled with higher accuracy. I recommend to plot CDR3 RMSD vs. highest sequence identity to the training set antibody to test this.

As I understand, the method does not require multiple sequence alignment as an input. I think this needs to be stated more explicitly.

We would like to thank the reviewers for their comments and believe the inclusion of their suggestions has greatly improved the manuscript. Below we give a point by point response, with the reviewers text in black, our response in blue and changes to the paper in red.

1 Reviewer

This study reports the development and benchmarking of ImmuneBuilder, which are deep learning protocols to model antibody, nanobody, and TCR structures. These algorithms are much faster than AlphaFold-Multimer, at least as accurate, and are available to the community via web server interfaces and Github. Additionally, sets of antibody structural models are made available by the authors on Zenodo. There are several aspects of the presentation of the results and methods that should be addressed, as noted in the comments below, but overall this is a very nice study that should be of considerable interest to the research community.

We thank the reviewer for the clear summary of our work and for their comments.

1.1 Comments

1. The authors note in the Results section (page 3): “ABodyBuilder2 is the most accurate method at predicting the structure of CDR-H3 (RMSD of 2.81), closely followed by AlphaFold-Multimer (RMSD of 2.90).” It is not clear whether this 0.09 improvement in mean RMSD is meaningful enough to say whether ABodyBuilder2 is truly better than AlphaFold-Multimer, or essentially the same in terms of accuracy. The authors should perform a statistical test (e.g. Wilcoxon signed rank on the individual CDRH3 RMSDs) to show whether the RMSDs from ABodyBuilder2 show a statistically significant improvement over AlphaFold-Multimer. Such a test would help to support any statements of superiority by the authors. If there is no significant difference or improvement, the authors should note that as well.

We thank the reviewer for pointing this out. We do not believe ABodyBuilder2 to be more accurate than AlphaFold-Multimer and tried to make this clear throughout the paper. We agree that line could be misinterpreted and have reworded that sentence to say the following:

ABodyBuilder2 and AlphaFold-Multimer are the most accurate methods at predicting the structure of CDR-H3 (RMSD of 2.81Å and 2.90Å respectively)

2. The authors do not seem to give the individual CDR RMSDs for their antibody, TCR, and nanobody test sets. Readers or users may want to know what the actual RMSDs were, versus just the reported mean values. This information should be provided as supplemental tables.

We agree with the reviewer that some readers may find this information useful and have added tables containing individual CDR RMSDs as supplemental tables. We have added the following sentence to appendix D.2 in the SI to indicate this.

The individual RMSDs for each method for each CDR are given as supplementary tables.

3. While it is understandable that the authors focus on antibody modeling performance in the main text and results, it would be helpful for readers to also be able to view the main performance results for TCRs and nanobodies, without needing to refer to the supplemental information. The two tables reporting accuracy performance (RMSDs) for nanobodies and TCRs should be moved from supplemental to the

main text and results.

We thank the reviewer for their suggestion. We have moved the accuracy performance tables for TCRs and nanobodies to the main text.

Tables 2 and 3 show the accuracy at predicting the structure of backbone atoms for TCRBuilder2 and NanoBodyBuilder2 respectively. We compare them to homology modelling methods (RepertoireBuilder [36] and TCRBuilder [26] for TCR modelling, and MOE [37] and ABodyBuilder [32] for nanobody modelling) and machine learning methods AlphaFold-Multimer [22] for TCRs and AlphaFold2 [21] for nanobodies. Full results for these methods and details on their respective test sets are given in the SI.

Method	CDR- α 1	CDR- α 2	CDR- α 3	Fw- α	CDR- β 1	CDR- β 2	CDR- β 3	Fw- β
TCRBuilder	1.60	1.31	2.89	0.87	0.99	0.90	3.12	0.81
RepertoireBuilder	1.35	1.00	2.64	0.75	0.86	1.59	2.77	1.05
AlphaFold-M	1.25	0.96	1.84	0.69	0.75	0.65	1.94	0.82
ABodyBuilder2	3.49	6.57	3.14	2.89	3.27	3.77	3.48	3.65
TCRBuilder2	1.34	0.93	1.85	0.90	0.74	0.63	1.93	0.67

Table 2 Comparison between TCRBuilder, RepertoireBuilder, AlphaFold-Multimer, ABodyBuilder2 and TCRBuilder2 at predicting the backbone atoms for each TCR chain. The mean RMSD to the crystal structure across the TCR test set for each of the six CDRs and framework (Fw) is shown. RMSDs are given in Angstroms (Å).

Method	CDR1	CDR2	CDR3	Fw
ABodyBuilder	2.96	2.08	5.08	1.09
MOE	2.67	1.99	4.90	1.19
AlphaFold2	2.08	1.35	3.44	0.82
NanoBodyBuilder2	1.98	1.37	2.89	0.79

Table 3 Comparison between ABodyBuilder, MOE, AlphaFold2 and NanoBodyBuilder2 at predicting the backbone atoms of nanobodies. The mean RMSD to the crystal structure across the nanobody test set for each of the three CDRs and framework (Fw) is shown. RMSDs are given in Angstroms (Å).

4. For the antibody results, the authors provide comparative results with available methods aside from AlphaFold-Multimer (i.e. IgFold and EquiFold), which is great. However, it would be nice if the authors likewise compared their nanobody and TCR modeling performance against at least one more method that was developed outside of their group and currently available to the community. That could include NanoNet, which was noted by the authors, and for TCRs, results for RepertoireBuilder or TCRmodel could be included, for example. This would help to provide a more comprehensive view on the reported performance for the test sets used by the authors.

We thank the reviewer their comment. For TCRs we have added a comparison with RepertoireBuilder to Table 2 of the main text and Table B2 in the SI. For nanobodies we already compare against MOE (a method for modelling nanobodies that was not developed in our group).

5. The Methods section (page 7) notes that “Finally, it was ensured that there were no structures with the same sequence in the test, training, and validation sets.” As written, it seems possible that two antibody structures with only one amino acid mismatch would possibly be in the training and test sets. The authors should more clearly note their nonredundancy criteria to ensure lack of train/test overlap (e.g. 99% identity, 95% identity) so that this is more clear to readers. If their training and test sets truly

have a relatively permissive identity cutoff (e.g. 99% identity threshold), the authors should comment on why this would not be a concern.

We thank the reviewer for bringing this up. The non-redundancy criteria used is as stated in the paper. There is one sequence in the test set with a sequence identity of over 99% to one in the training set. However, the one amino acid mismatch is an insertion in CDR-H3 that significantly affects the structure.

As suggested by reviewer 2 we have added a plot comparing CDR-H3 RMSD vs. highest sequence identity to the training set to the SI. Hopefully, this clarifies why we consider the non-redundancy cutoff used to be sufficient. We have added the following sentence to the Methods section of main text:

A comparison of the maximum sequence identity to the training set against CDR-H3 RMSD for each Fv in the test set is shown in SI Figure D3.

And the following paragraph to the SI:

Only antibody structures with an identical heavy and light chain sequence to those in the training set were excluded from the benchmark set. For each antibody in our benchmark, the most similar antibody in our training set has a sequence identity with values ranging from 62% to 99.5% (the latter having a single insertion in CDR-H3). Figure D3 shows that having highly identical sequences in the training set does not necessarily improve the models ability to accurately predict CDR-H3.

SI Fig. D3 CDR-H3 RMSD versus maximum sequence identity to the training set for ABodyBuilder2.

6. It is reported by the authors that AlphaFold-Multimer was run without the use of templates for the antibody modeling. It is not clear why templates were not used, particularly as it is possible to set a template date cutoff to avoid overlap with the test set, and it is theoretically possible that templates

may help with the AlphaFold-Multimer modeling. The authors should run AlphaFold-Multimer for their antibody test set, allowing templates with the appropriate date cutoff to avoid recent structures that would overlap with their test set, and report that performance. This would provide a more useful comparator for readers, who would potentially run AlphaFold-Multimer’s default protocol, which does include templates, on prospective modeling targets.

We thank the reviewer for pointing this out. We have rerun AlphaFold-Multimer on the antibody test set using templates and shown that it does not significantly change the results. We have modified the following sentences in the Methods section of the main text:

AlphaFold-Multimer was run using the freely available version of the code [22]. It was run using the weights from version 2.2 and without the use of templates. The effect of templates on antibody structure prediction is shown in SI Table D3.

And the following paragraphs to the SI:

Throughout the paper we compare our methods against AlphaFold2 without the use of templates. In Table D3 we compare the effect of using templates has on AlphaFold-Multimer predictions for the antibody benchmark. We only allow AlphaFold-Multimer to use templates from structures released before the 1st of January 2022 to ensure it does not use any structures in our test set.

Method	CDR-H1	CDR-H2	CDR-H3	Fw-H	CDR-L1	CDR-L2	CDR-L3	Fw-L
AFM (no templates)	0.86	0.68	2.90	0.55	0.47	0.40	0.83	0.54
AFM (templates)	0.84	0.67	2.88	0.55	0.49	0.42	0.81	0.57
AB2	0.85	0.78	2.81	0.54	0.46	0.44	0.87	0.57

SI Table D3 Comparison of performance when running AlphaFold-Multimer with or without templates for the antibody benchmark set.

As can be seen from Table D3, the use of templates results in no significant improvement to the prediction accuracy of AlphaFold-Multimer on antibodies. However, we found that antibody structures generated using AlphaFold-Multimer with templates had a higher number of stereochemical errors. For the 34 antibodies in the benchmark set, AlphaFold-Multimer models were found to have two clashes, three unphysical peptide bonds and three D-amino acids. The non-template version of AlphaFold-Multimer generates none of these without any significant loss in accuracy, so it was used in all our benchmarks.

7. On the ImmuneBuilder site, a TCR model was successfully generated (which is nice), but selecting “Prediction error” under “Annotation options” for the TCR model made the model disappear in the viewer. However, the prediction error is visible for the sample TCR output on the web server. The authors should check this, to make sure that it that aspect of the model output page is functioning properly, and if not, it should be fixed.

We thank the reviewer for bringing this up and apologise for the inconvenience. There was a bug in our web server that has since been fixed.

8. Page 2. “sitting between two Ig domains” does not seem to reflect the antigen binding of TCRs and antibodies, as the CDR loops are not actually between the two chains (e.g. within the VH-VL interface), they just form a contiguous surface formed by both domains. Thus the authors should consider re-wording that sentence.

We thank the reviewer for pointing this out. We have changed the wording from "between" to "across".

All three of these immune proteins are built up from immunoglobulin (Ig) domains with the binding site either sitting across two Ig domains in the case of antibodies (VH and VL) and TCRs ($V\alpha$ and $V\beta$), or being found at the tip of one Ig domain (VHH), in the case of nanobodies.

9. The authors should include a statement or paragraph noting any shortcomings, failures, or areas of potential improvement for these methods.

We thank the reviewer for indicating this. We have added a paragraph in the discussion mentioning shortcomings and suggesting areas where improvements are needed.

The comparison with homology modelling methods, such as ABodyBuilder, shows the benefits that deep learning has brought to the field of antibody structure predictions. However, all methods still struggle to accurately predict the conformation of CDR-H3, suggesting that models capable of predicting multiple conformations may be required to accurately capture this loop. Deep learning methods also still struggle to consistently predict physically plausible structures. This challenge can be addressed by using physics-based methods, such as restrained energy minimisation, but for fast methods like ABodyBuilder2 this significantly increases computational cost.

2 Reviewer

The manuscript presents a set of deep learning models for predicting structures of antibodies, nanobodies and T-cell receptors. The architecture of the models is inspired by AlphaFold2, adjustments to immune receptors are made. The method is trained and validated using existing structures and achieving state-of-the-art performance. Overall, the method is an excellent addition to the existing structure prediction approaches for immune receptors. The authors provide a notebook for predicting the structures using a colab notebook and a webserver. The source code is also available, although I did not find the training part.

We thank the reviewer for the clear summary of our work and for their comments.

2.1 Comments

1. Only mean RMSD values are reported for the test set. It would be useful to add plots with distributions, such as swarmplots, especially for CDR3. Do the different methods fail on the same test cases or different ones? Is it possible to include scatterplots of CDR3 RMSDs for comparison to additional methods? Highlight cases where ABodyBuilder2 is successful and where it fails?

We agree with the reviewer that some readers may want this information and have added scatter plots comparing CDR-H3 RMSDs between each pair of methods to the SI (SI Fig. D4), and have added tables containing individual CDR RMSDs as supplemental tables. We have added the following sentence in the Results section of main text to indicate this:

A comparison of the CDR-H3 RMSD for each individual structure in the test set between each pair of methods is shown in SI Figure D4.

We have also added the following paragraph and figure to the SI:

In Figure D4, the CDR-H3 RMSD for each antibody in our test set is compared for each pair of benchmarked methods. In the majority of cases, ABodyBuilder2 is consistently better for most antibodies in the benchmark. The exception to this is AlphaFold-Multimer, where there appears to be a significant number of antibodies for which AlphaFold-Multimer significantly outperforms ABodyBuilder and vice versa. It hence may be beneficial for some applications to combine the predictions from both methods. The individual RMSDs for each method for each CDR are given as supplementary tables.

2. Splitting the data to train/test – only identical sequences in the validation and train sets were removed from the training set. This is suboptimal and most likely test set sequences that have immune receptors with high sequence identity in the training set are modeled with higher accuracy. I recommend to plot CDR3 RMSD vs. highest sequence identity to the training set antibody to test this.

We thank the reviewer for their comment. As suggested we have added a plot comparing CDR-H3 RMSD vs. highest sequence identity to the training set to the SI. (SI Fig. D3) We have added the following sentence to the Methods section of main text:

A comparison of the maximum sequence identity to the training set against CDR-H3 RMSD for each Fv in the test set is shown in SI Figure D3.

And the following paragraph to the SI:

Only antibody structures with an identical heavy and light chain sequence to those in the training set were excluded from the benchmark set. For each antibody in our benchmark, the most similar antibody in our training set has a sequence identity with values ranging from 62% to 99.5% (the latter having a single insertion in CDR-H3). Figure D3 shows that having highly identical sequences in the training set does not necessarily improve the model's ability to accurately predict CDR-H3.

SI Fig. D4 Comparison of CDR-H3 RMSDs for each antibody in the test set for each of the benchmarked structure prediction methods. On the diagonal, a histogram showing the distribution of CDR-H3 RMSDs for each method is shown. Off diagonal, scatter plots comparing the ability of each method to model each CDR-H3 in the benchmark is shown. RMSDs are given in Angstroms (\AA)

SI Fig. D3 CDR-H3 RMSD versus maximum sequence identity to the training set for ABodyBuilder2.

3. As I understand, the method does not require multiple sequence alignment as an input. I think this needs to be stated more explicitly.

The reviewer is correct in that the method does not need multiple sequence alignments as input. This is one of the benefits of our method over AlphaFold2. We have added the following line to the Discussion section to make this more clear.

ImmuneBuilder can produce structures of antibodies and TCRs with accuracy comparable to AlphaFold-Multimer while being over a hundred times faster and without the need for large sequence databases or multiple sequence alignments.

REVIEWERS' COMMENTS:

Reviewer #1 (Remarks to the Author):

The authors have addressed all comments.

Reviewer #2 (Remarks to the Author):

The authors revised the manuscript according to comments.